# Towards Unified Multi-Modal Personalization: Large Vision-Language Models for Generative Recommendation and Beyond

**Tianxin Wei[1][§], Bowen Jin[1][§], Ruirui Li[2], Hansi Zeng[3][§], Zhengyang Wang[2], Jianhui Sun[4][§],**
**Qingyu Yin[2], Hanqing Lu[2], Suhang Wang[5], Jingrui He[1], Xianfeng Tang[2]**
[1]University of Illinois Urbana-Champaign  [2]Amazon  [3]University of Massachusetts Amherst
[4]University of Virginia  [5]The Pennsylvania State University
{twei10,bowenj4,jingrui}@illinois.edu, hzeng@cs.umass.edu, js9gu@virginia.edu
szw494@psu.edu, {ruirul,zhengywa,qingyy,luhanqin,xianft}@amazon.com

## Abstract

Developing a unified model that can effectively harness heterogeneous resources and respond to a wide range of personalized needs has been a longstanding community aspiration. Our daily choices, especially in domains like fashion and retail, are substantially shaped by multi-modal data, such as pictures and textual descriptions. The vision and language modalities not only offer intuitive guidance but also cater to personalized user preferences. However, the predominant personalization approaches mainly focus on the ID or text-based recommendation problem, failing to comprehend the information spanning various tasks or modalities. In this paper, our goal is to establish a Unified paradigm for Multi-modal Personalization systems (UniMP), which effectively leverages multi-modal data while eliminating the complexities associated with task- and modality-specific customization. We argue that the advancements in foundational generative modeling have provided the flexibility and effectiveness necessary to achieve the objective. In light of this, we develop a generic and extensible personalization generative framework, that can handle a wide range of personalized needs including item recommendation, product search, preference prediction, explanation generation, and further user-guided image generation. Our methodology enhances the capabilities of foundational language models for personalized tasks by seamlessly ingesting interleaved vision-language user history information, ensuring a more precise and customized experience for users. To train and evaluate the proposed multi-modal personalized tasks, we also introduce a novel and comprehensive benchmark covering a variety of user requirements. Our experiments on the real-world benchmark showcase the model's potential, outperforming competitive methods specialized for each task.

## 1 Introduction

With rapid growth, personalization systems have emerged as a key factor in meeting the user's expectations for tailored experiences that align with their unique needs and preferences. In today's digitally driven landscape, individuals engage with diverse data types, such as ratings, images, descriptions, and prices, especially in domains like fashion and retail (Kang et al., 2017; Hwangbo et al., 2018) where visuals and text are essential for decision-making. Given the profound influence of these multi-modal stimuli, there exists a pressing need for systems that can seamlessly integrate and harness these diverse data streams for improved personalization. Moreover, users often require a variety of information systems to meet their different needs. For example, e-commerce platforms commonly offer both search and recommendation functionalities to users (Luo et al., 2022). The combination of accurate preference prediction, persuasive recommendation explanation and intuitive visual guidance could further enhance the user experience. In response to these evolving demands, there's an emerging imperative to develop the *Unified Multi-modal Personalization system*

---

§Work was done while Tianxin, Bowen, Hansi, Jianhui were interning at Amazon.

(`UniMP`). Such system aims to cater to individualized preferences that are often hidden in the complex interplay between different modalities and data types, streamlining engineering processes and uniting research efforts working on various aspects of personalized user modeling.

Decades of research have been dedicated to bolster the personalization capability of models for each specialized scenario. Standard and well-established methods include matrix factorization and neural networks (CNN/RNN/Self-attentive models). They convert user consumption logs into sequences of item IDs and create embedding tables for encoding. In order to augment the context information, some approaches also incorporate side features such as item categories (Zhou et al., 2020; Hou et al., 2022) into the framework. Within a carefully designed network, the final representation can then be employed for specific tasks, such as matching the items that are likely to pique user interests.

Despite their effectiveness, traditional methods suffer from several key flaws. First, these models struggle to process multi-modal personalization data from user histories. User histories often comprise a mixture of visual, textual, Product ID and tabular data, arranged in arbitrary sequences. Effectively utilizing such heterogenous data input poses significant challenges to the models. Secondly, while some research works (Geng et al., 2023) employ pre-trained models like BERT (Devlin et al., 2018), ResNet (He et al., 2016), or ViT (Dosovitskiy et al., 2020) to derive textual or visual representations, these models are usually trained on a general corpus, often differing in language or visual contexts. In light of this, the model cannot discern the importance of various textual attributes and visual elements, hindering its ability to understand fine-grained user preferences for personalization tasks. Moreover, loosely integrating the multi-modal representation fails to grasp the complex interactions of the cross-modality data during the user modeling process. Lastly, it often requires careful per-task network architecture design and hyper-parameter tuning. The process is resource-intensive and also hinders knowledge transfer across various tasks. Even the recent works (Geng et al., 2023; 2022) address only a relatively narrow range of use cases and don't tackle the multi-task learning optimization dilemma.

Recently, pre-trained generative models with instructional tuning (Chung et al., 2022; Liu et al., 2023) have demonstrated remarkable generalization abilities on a variety of tasks. In this project, we propose the Unified Multi-modal Personalization system (`UniMP`) to utilize and extend the capabilities of the generative language model, facilitating multi-modal understanding and enhancing the system's personalization capabilities to address the aforementioned limitations. First, instead of restricting to specific data types, `UniMP` introduces a unified data format which is flexible enough to seamlessly incorporate various types of user history information, including images, texts, IDs, etc. Each user consumed items can include varying attributes as contextual elements. Second, to better understand multi-modal information interactions and their sequential transitions, we propose an innovative user modeling architecture based on Alayrac et al. (2022), in which a vision model extracts the visual elements and a language model is responsible for reasoning and generation based on user history. Visual information of each item is conditioned on the corresponding textual representation throughout the layers of the language model, allowing for fine-grained multi-modal information extraction and alignment. Moreover, inspired by prior examination of information retrieval and filtering systems (Belkin & Croft, 1992), which suggests that joint modeling and optimization can lead to higher generalization, we further broaden the hypothesis to encompass multiple multi-modal tasks. Leveraging the power of foundation models, we propose to conceptualize each multi-modal task as an instruction, integrating multiple personalization task learning within a token generation framework to learn transferable knowledge. To address the dilemma of the multi-task optimization, we introduce token-level re-weighting to enable fine-grained prioritization of samples across diverse tasks, and employ context reconstruction to improve training efficacy. Notably, our flexible framework not only accommodates multi-modal input but also supports multi-modal output generation. This is achieved by quantizing images into visually-coherent tokens Esser et al. (2021).

To evaluate `UniMP`, we present a novel and comprehensive benchmark tailored for a variety of user requirements. The proposed `UniMP` facilitates multi-task multi-modal personalized learning, and efficient fine-tuning on each specific task can further enhance the overall performance. Our contribution can be summarized as follows:

- We introduce a unified data format to seamlessly ingest various types of user history information. Our flexible framework not only accommodates multi-modal user inputs but also facilitates multi-modal output generation to fulfill personal needs.

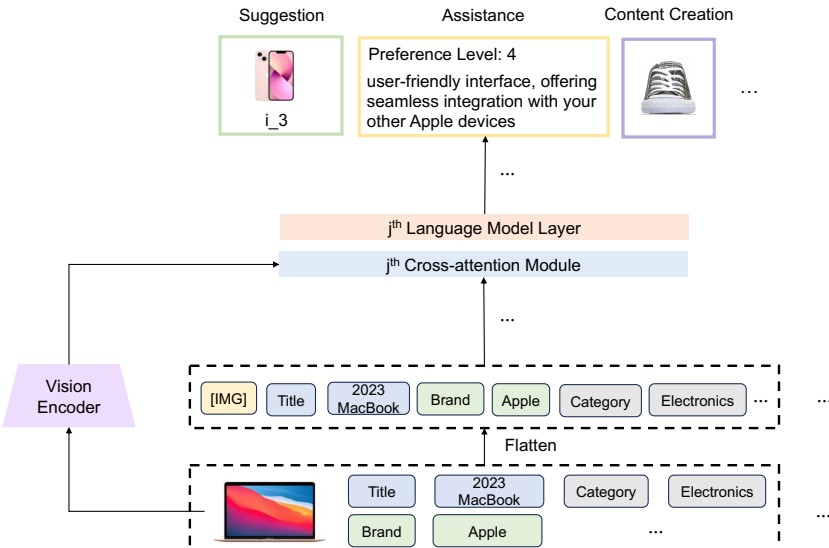

Figure 1: Our proposed `UniMP` framework operates as follows: Item contextual data is streamlined into a user sequence, which is then processed through fine-grained cross-modal fusion. Depending on the instructions, the output is tailored to produce diverse response types.

- Our innovative user modeling architecture is designed for fine-grained multi-modal information extraction and alignment, ensuring precise user preference predictions. We also present an effective multi-task optimization approach that bolsters the generalization capabilities of `UniMP`.
- We propose to unify and benchmark representative personalization tasks to evaluate diverse personal needs and heterogeneous knowledge sources. Extensive experiments are conducted to show the effectiveness of `UniMP`. Results show the model can outperform competitive methods specialized for each task and substantially improve the model's transferability.

## 2 PROPOSED METHOD

In this section, we present our proposed `UniMP`. We first introduce how to seamlessly ingest heterogeneous user history information into a large language model. Subsequently, we illustrate the architecture of the multi-modal user modeling framework, facilitating fine-grained user preference understanding. Lastly, we propose to formulate personalization tasks as multi-modal prompts, effectively leveraging a generative language model to build a general-purpose e-commerce system. To overcome challenges in multi-task optimization, we employ context reconstruction to improve training efficacy and introduce token-level re-weighting to automatically prioritize optimization across different contexts, samples and tasks. The overview of our proposed model is shown in Figure 1.

### 2.1 INCORPORATE HETEROGENEOUS USER HISTORY INFORMATION

For the multi-modal personalized tasks, we are given a user's interaction history $u_h = \{I_1, I_2, \ldots, I_n\}$ in temporal order where $I_e$ denotes the contextual information of item $e$ and $n$ is the length of $u_h$. In many prior works on sequential recommendation, each interacted item $i$ is solely associated with a unique item ID. In real-world scenario, $I_e$ of item $e$ in $u_h$ contains rich heterogenous information including image, category, brand, description, price, etc. Based on $u_h$, we seek to introduce a unified input format to harness the multi-modal heterogeneous content. Toward this goal, we construct an attribute dictionary $D_e$ for each corresponding item $e$, which consists of heterogenous key-value attribute pairs $I_e = D_e = \{(\text{"image"}, x), (k_1, v_1), (k_2, v_2), \ldots, (k_m, v_m)\}$. Here $k$ denotes an attribute name (e.g., Brand), $v$ is the corresponding value (e.g., Apple) and $x$ is the visual input of the item. $k$ and $v$ are both described by natural languages and contain words $(k, v) = \{w_1^k, \ldots, w_c^k, w_1^v, \ldots, w_c^v\}$, where $w^k$ and $w^v$ are words of $k$ and $v$ from a shared vocabulary in the language model and $c$ denotes the truncated length of text. The unified attribute dictionary

$D_e$ can include all kinds of available item information such as images, titles, descriptions, colors, etc.

In order to harness the dictionary $D_e$ with the model, we flatten key-value attribute pairs into

$$i_e = \text{Flatten}(I_e) = \{[\text{IMG}], k_1, v_1, k_2, v_2, \ldots, k_m, v_m, [\text{EOC}]\} \tag{1}$$

A special token [IMG] in plain text is inserted into the item sequence to denote the location of the item image, setting the stage for subsequent processing of $x$. Following Alayrac et al. (2022), another special token [EOC] (end of chunk) is used to denote the end of the textual and visual description of each item. The sequences of items interacted by the user are combined to form the user sentence denoted as

$$u_n = \{[\text{CLS}], i_1, i_2, \ldots, i_n\} \tag{2}$$

We then replace the user history with the user sentence, ensuring no loss of information, and incorporate a special token [CLS] at the beginning of the sequence. Notably, unlike previous personalized systems (Geng et al., 2022; Li et al., 2023b) using only text or item IDs, in this study, we aim to handle multi-modal heterogeneous user sequence. This sequence including visual information isn't immediately compatible with the language model.

## 2.2 FINE-GRAINED MULTI-MODAL USER MODELING

To solve the problem, here we present a multi-modal model designed to process the multi-modal user sequence consisting of both text and interleaved images. The key architectural components shown in Figure 1 are chosen to leverage pre-trained vision and language models (LMs) and bridge them effectively. To effectively process the modalities beyond text. The visual input is fed into the pre-trained vision encoder $f_v$ to obtain the features. Following (Alayrac et al., 2022), the vision sampler receives visual features from the vision encoder and design learnable query vectors to attend to the features and output a fixed number of visual embeddings $x_v \in \mathbb{R}^{H \times C \times d}$ with the same hidden dimension as the text encoder. $H$ is the number of images in the user sequence, $C$ is the number of extracted visual embeddings and $d$ is the hidden dimension. The user sentence in plain text form is simultaneously fed into the pretrained language model. These visual tokens are then used to condition the frozen LM in a fine-grained manner using cross-attention mechanism. For the cross-attended layer $j$, denote the textual token embedding of the user sentence as $\hat{x}_t^j \in \mathbb{R}^{N \times d}$ where $N$ is the number of tokens in the sequence. The cross-attention mechanism can be described as:

$$\tilde{x}_t^j = \text{Cross\_Att}(Q = \hat{x}_t^j, KV = x_v, M) = \text{Softmax}\left(\frac{QK^T}{\sqrt{d}} \cdot M\right) V \tag{3}$$

$$x_t^j = g \cdot \tilde{x}_t^j + \hat{x}_t^j \tag{4}$$

where the query vector is the output textual embedding at $j$-th layer, and the key as well as the value vectors are the extracted vision embeddings. The mask matrix, denoted as $M$, is constructed using special tokens to indicate the correspondence between textual tokens and images. Its purpose is to ensure that the vision information exclusively conditions on the corresponding textual information during cross-modal fusion. Subsequently, the fused information undergoes a learnable gating mechanism denoted as $g$ (Hochreiter & Schmidhuber, 1997) and a residual connection (He et al., 2016). These components enable flexible control over the integration of visual information. The architecture offers an expressive and fine-grained way for the LM to incorporate supplementary historical visual information for user understanding.

## 2.3 INTEGRATION OF PERSONALIZATION TASKS

`UniMP` proposes to integrate multiple personalization tasks within a token generation framework, leveraging the power of foundation models. Based on the requirement of various tasks, we can formulate each multi-modal personalized task as the following next-token prediction objective:

$$\min - \log p(y \mid u) = -\log p(y_{>n} \mid u_{\leq n} = \{u_n, T\}) \tag{5}$$

where $y_{>n}$ is the prediction token target based on the input user and task information, $T$ is the task description and $u_{\leq n}$ is the set of preceding tokens of the user history sequence with length $n$. For example, recommendation task is to predict the relevant item IDs that the user will be interested

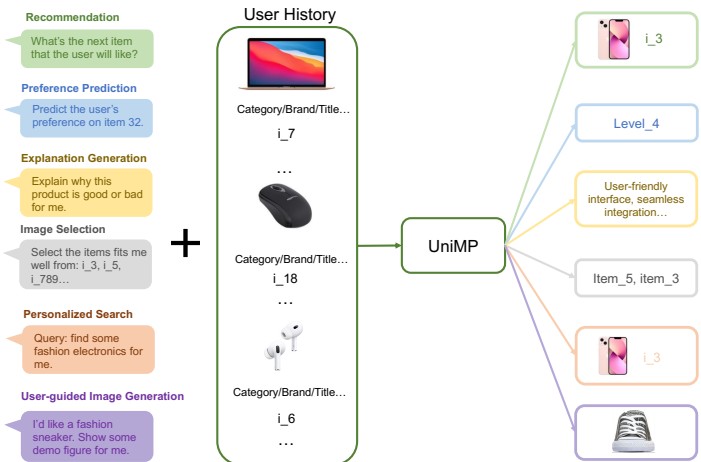

Figure 2: Through multi-task, multi-modal instruction tuning, the model can adapt to a range of user requirements. By altering the instructions, it can generate diverse responses to suit user needs. For in-depth and exact information on task instructions and input formats, please see Appendix A.2.

in. To unify and optimize multiple tasks together, each task can be described by an instruction $T$. For example, the recommendation task can be described as: "What's the next item that the user may like?" By conceptualizing all the tasks as textual instructions, and combining it with the user history information, we can unify multiple tasks together with the unified instruction format. The framework is shown in Figure 2. An intuitive solution is continuous learning from all tasks. However, we observe it is hard to design the order of task learning and the prior knowledge will easily be forgetton. Thus, we formulate the final objective as joint multi-task learning:

$$\min \sum_{a=1}^{A} \lambda_a \cdot \mathbb{E}_{(u,y)\sim\mathcal{T}_a} \left[ -\sum_{\ell=1}^{L} \log p\left(y_{>n} \mid u_{\leq n} = \{u_n, T_a\}\right) \right] \quad (6)$$

where $\lambda_a$ is the trade-off loss coefficient for Task $a$, $A$ is the number of tasks, $T_a$ and $\mathcal{T}_a$ represents the task description and task instruction data of $a$. Though the above loss looks promising, the naive multi-task learning faces several problems. First, the prediction objective based on multi-sample/image user history does not match with the pre-trained task which performs single-sample/image prediction. This discrepancy raises concerns about the model's ability to effectively leverage the pre-trained knowledge when it is required to make prediction based on whole user history. Second, it is difficult to identify the importance of various tasks. Manually tuning $\lambda_a$ is labor intensive and only achieves coarse-grained weight adjustment, leading to sub-optimal performance. To address the above problems, we propose the following strategies.

**Context Reconstruction.** Our previous model only optimize the loss for the task response $\ell_{\text{task}}$ based on complete user history $u_{\leq n}$ of multiple samples. Our goal is to reconcile this by integrating modifications that align the prediction objective more closely with the pre-training task of sample-level prediction. Thus, we propose optimizing the context reconstruction loss terms jointly. This can be easily included in the optimization without much extra computational cost, and boosts training effectiveness. The revised objective is given as:

$$\ell = \ell_{\text{task}} + \ell_{\text{context}} = -\log p(y_{>n} \mid u_{\leq n} = \{u_n, T\}) - \log p(u_n = \{i_1, i_2, ...\}) \quad (7)$$

where the two loss corresponds to the token prediction loss for the task response and context of each historical item.

**Token-level Re-weighting.** Then, to prioritize the samples from different tasks and contexts, we propose to reshape the loss function to re-weight the token prediction loss. Inspired by (Lin et al., 2017), we down-weight easy tokens and thus focus training on hard tokens by incorporating a modulating factor $(1 - p)^\gamma$ to the likelihood, where $\gamma$ is the tunable focusing parameter and $p$ is the standard token prediction likelihood. The process of re-weighting becomes particularly meaningful

when optimizing both the task loss and context loss concurrently, given that contextual information tends to be more predictable. Without proper re-weighting, the model may be susceptible to converging into local minima during training. The new token probability $p_w$ is defined as:

$$p_w = (1 - p)^\gamma \log p \tag{8}$$

The re-weighted likelihood offers a fine-grained prioritization of token loss optimization, balancing the contributions of different tasks and samples, thereby enhancing effectiveness.

**Retrieval-augmented Generation beyond Text.** Beyond text, our framework can also be adapted to generate other data types, such as images. Building on (Esser et al., 2021), images can be quantized into sequences of visually coherent tokens, which the transformer can then learn to generate. These tokens can subsequently be translated back to the original images using the CNN decoder from (Esser et al., 2021). Such quantization capitalizes on the language model's prowess in long-term modeling of realistic images. Here we formulate the task of User-guided Image Generation, aimed at real-time generation of personalized image content. This facilitates users in locating and exploring intriguing items, enhancing their experience and addressing varied informational needs. However, generating images with precision can be largely hampered by the interference of noisy historical data. To address this, we've introduced an intermediate search phase to identify relevant items. These items then help guide the accurate generation of image tokens, as delineated below:

$$S = \text{Beam}[p(y_{>n} \mid u_{\leq n} = \{u_n, T_s\})], \quad G = \text{Beam}[p(y_{>n} \mid u_{\leq n} = \{S, T_g\})] \tag{9}$$

where $T_s$ denotes the retrieval task instruction based on the user query, $\text{Beam}$ represents generation process from beam search algorithm, $S$ is the set of items retrieved by the model and $T_g$, $G$ are image generation instruction and output tokens. The tokens will be fed into quantization decoder to get the actual image.

## 3 EXPERIMENTS

In this section, we empirically evaluate the performance of our proposed `UniMP`, intending to answer the following research questions:

- **RQ1:** Does `UniMP` enable effective learning for various multi-modal personalization tasks?
- **RQ2:** How do different components in our framework contribute to the performance?
- **RQ3:** How well does our proposed `UniMP` generalize across different scenarios, such as with new users and in unfamiliar domains?
- **RQ4:** Can `UniMP` generate personalized content to user's query?
- **RQ5:** How robust is `UniMP` when faced with noisy multi-modal input scenarios? (Appendix A.3)

### 3.1 EXPERIMENTAL SETUP

**Datasets.** We assessed the model's personalization capabilities with multi-modal input by conducting experiments on several domains from the publicly available Amazon review dataset (He & McAuley, 2016a; Ni et al., 2019). These domains include: Amazon-Baby, Amazon-Beauty, Amazon-Clothing, Amazon-Grocery, Amazon-Sports, Amazon-Toys, and Amazon-Office. Of these, Amazon-Office was reserved for testing the model's generalizability to new domains, while the others served as training data and in-domain evaluation. We combined all the datasets to create a more realistic and comprehensive dataset. Only users and items with at least five interactions were included for training. For each user, the latest interaction served as the test sample, the second-to-last as the validation sample, and all earlier interactions were used for training. Detailed information about the dataset is provided in Table 5 in Appendix A.1. To provide a comprehensive evaluation, we introduced a new benchmark that encompasses several multi-modal tasks, showcasing the multi-modal system's personalization potential. These tasks encompass personalized image generation, multi-modal explanation, and multi-modal personalized search. Collectively, these tasks represent a wide range of current user needs.

**Implementation Details.** For visual encoder, we utilize the CLIP ViT-L model (Radford et al., 2021). For the pre-trained large language model, we adopt the instruction-tuned 3B model (Together.ai, 2023). During model training, by default, we train our model for 10 epochs with a learning

rate of 2e-4 with AdamW optimizer (Loshchilov & Hutter, 2017) and batch size 24 on an 8 A100 40G machine. We utilize the cosine learning scheduler to achieve better convergence. We compare with the representative and state-of-the-art baselines of each task. For recommendation, we split the baselines into four categories: general recommenders, sequential recommenders, generative model-based recommenders and multi-modal recommenders. More details about implementation, tasks and baselines can be found in Appendix A.2.

Table 1: Experimental Comparisons on Product Recommendation.

|  | HR@3 | NDCG@3 | MRR@3 | HR@5 | NDCG@5 | MRR@5 |
|---|---|---|---|---|---|---|
| MF | 0.0105 | 0.0077 | 0.0065 | 0.0165 | 0.0093 | 0.0078 |
| MACR | 0.0110 | 0.0080 | 0.0068 | 0.0170 | 0.0105 | 0.0091 |
| LightGCN | 0.0142 | 0.0103 | 0.0088 | 0.0206 | 0.0129 | 0.0094 |
| UltraGCN | 0.0151 | 0.0111 | 0.0095 | 0.0215 | 0.0134 | 0.0102 |
| HGN | 0.0167 | 0.0113 | 0.0113 | 0.0231 | 0.0153 | 0.0117 |
| GRU4Rec | 0.0132 | 0.0101 | 0.0086 | 0.0201 | 0.0128 | 0.0096 |
| SASRec | 0.0189 | 0.0124 | 0.0102 | 0.0276 | 0.0175 | 0.0126 |
| $S^3$-Rec | 0.0205 | 0.0149 | 0.0133 | 0.0311 | 0.0193 | 0.0156 |
| BERT4Rec | 0.0121 | 0.0092 | 0.0079 | 0.0198 | 0.0127 | 0.0105 |
| UniSRec | 0.0201 | 0.0146 | 0.0135 | 0.0281 | 0.0191 | 0.0158 |
| P5 | 0.0086 | 0.0056 | 0.0042 | 0.0124 | 0.0074 | 0.0061 |
| VIP5$^+$ | 0.0175 | 0.0125 | 0.0108 | 0.0262 | 0.0163 | 0.0127 |
| VBPR | 0.0114 | 0.0084 | 0.0071 | 0.0181 | 0.0102 | 0.0086 |
| CausalRec | 0.0143 | 0.0105 | 0.0088 | 0.0229 | 0.0146 | 0.0121 |
| MMGCL | 0.0151 | 0.0112 | 0.0095 | 0.0241 | 0.0159 | 0.0130 |
| MMSSL | 0.0181 | 0.0132 | 0.0112 | 0.0281 | 0.0194 | 0.0164 |
| UniMP (Ours) | **0.0248** | **0.0194** | **0.0176** | **0.0337** | **0.0231** | **0.0196** |

Table 2: Personalized Task Performance Evaluation.

|  | BLEU | ROUGH | METEOR | BERTSCORE |
|---|---|---|---|---|
| P5 | 0.0822 | 0.0624 | 0.0321 | 0.723 |
| VIP5$^+$ | 0.0967 | 0.0723 | 0.0383 | 0.756 |
| MRG | 0.1101 | 0.0754 | 0.0466 | 0.812 |
| UniMP (Ours) | **0.1423** | **0.1080** | **0.0678** | **0.853** |

(a) Personalized Explaination Generation

|  | MAE | RMSE |
|---|---|---|
| P5 | 1.1234 | 1.8756 |
| VIP5$^+$ | 0.9452 | 1.5537 |
| SANCL | 0.9122 | 1.5781 |
| UniMP (Ours) | **0.7049** | **1.1651** |

(b) Personalized Preference Prediction

|  | Recall | Precision | F1 |
|---|---|---|---|
| P5 | 0.6882 | 0.6111 | 0.6326 |
| VIP5$^+$ | 0.7625 | 0.6952 | 0.7044 |
| $S^3$-Rec | 0.7056 | 0.6443 | 0.6572 |
| MMSSL | 0.7431 | 0.6721 | 0.6955 |
| UniMP (Ours) | **0.9665** | **0.8877** | **0.9111** |

(c) Personalized Multi-modal Selection

|  | HR@5 | NDCG@5 | MRR@5 |
|---|---|---|---|
| BM25 | 0.0513 | 0.0378 | 0.0314 |
| HEM | 0.0654 | 0.0474 | 0.0368 |
| ZAM | 0.0842 | 0.0615 | 0.0542 |
| RTM | 0.0981 | 0.0706 | 0.0587 |
| UniMP (Ours) | **0.1769** | **0.1309** | **0.1158** |

(d) Personalized Multi-modal Search

## 3.2 Results (RQ1)

Performance comparisons across various personalization tasks for different methods can be found in Tables 1, 2a, 2b, 2c, and 2d. Several key observations emerge from these tables: Our method substantially and consistently surpasses other competitive baselines across all tasks, outperforming those specifically designed for each respective task. Notably, in the challenging recommendation scenario where no prior information is available, our approach still outperforms the baselines by a considerable margin. This performance difference widens further for other tasks where additional information is provided by user. An intriguing pattern we noticed is the method's enhanced performance at higher positions. Specifically, the improvement in HR@3 and NDCG@3 over baselines is substantially greater than the improvement seen in HR@5/NDCG@5.

## 3.3 Ablation Study (RQ2)

In Table 3, we present an ablation study for `UniMP`. The following notations are used: "Vision": Utilization of vision information during the user modeling process. "Fine-grained": Implementation of the layer-wise cross-attention mechanism, rather than a simplistic integration of visual and

Table 3: Ablation study of `UniMP` on Recommendation Task.

|  | HR@5 | NDCG@5 | MRR@5 |
|---|---|---|---|
| w/o Vision | 0.0284 | 0.0205 | 0.0169 |
| w/o Fine-grained | 0.0257 | 0.0192 | 0.0161 |
| w/o Context & Rew | 0.0311 | 0.0214 | 0.0184 |
| w/o Context | 0.0318 | 0.0225 | 0.0188 |
| w/o Rew | 0.0302 | 0.0212 | 0.0178 |
| UniMP (Ours) | **0.0337** | **0.0231** | **0.0196** |

textual embeddings. "Context": Incorporation of the contextual reconstruction component. "Rew": Token-level re-weighting mechanism. The results indicate that each component plays a crucial role in the model's performance. A drop in performance is observed when any of these components are removed. Particularly, the vision information and fine-grained modeling emerge as pivotal factors in achieving optimal performance. Moreover, we note that excluding the re-weighting mechanism results in a lower performance than when both contextual reconstruction and re-weighting are removed. This observation underscores the importance of re-weighting, especially when optimizing both task and context losses, given the predictability of contextual information.

## 3.4 GENERALIZATION ABILITY (RQ3)

Table 4: Experiments of Generalization Ability of `UniMP`.

| | HR@5 | NDCG@5 | MRR@5 | | HR@5 | NDCG@5 | MRR@5 |
|---|---|---|---|---|---|---|---|
| P5 | 0.0119 | 0.0079 | 0.0062 | P5 | 0.0141 | 0.0081 | 0.0062 |
| VIP5 | 0.0217 | 0.0143 | 0.0124 | VIP5 | 0.0235 | 0.0167 | 0.0148 |
| S$^3$-Rec | 0.0228 | 0.0140 | 0.0123 | S$^3$-Rec | 0.0226 | 0.0152 | 0.0128 |
| UniSRec | 0.0245 | 0.0152 | 0.0131 | UniSRec | 0.0278 | 0.0184 | 0.0175 |
| UniMP (Ours) | **0.0364** | **0.0263** | **0.0228** | UniMP (Ours) | **0.0433** | **0.0289** | **0.0242** |
| (a) New User Recommendation | | | | (b) New Domain Recommendation | | | |

In Table 4a and Table 4b, we assess the generalization capabilities of different methods when evaluated on new users and new domain that are not present in the training set. Note that most baselines can not handle the new users introduced after training as they need to optimize the new user embeddings from scratch. It is evident that UniMP reaps greater advantages in the transfer learning contexts, primarily attributed to its efficient exploitation of heterogeneous user histories.

## 3.5 VISUALIZATION OF CONTENT GENERATION. (RQ4)

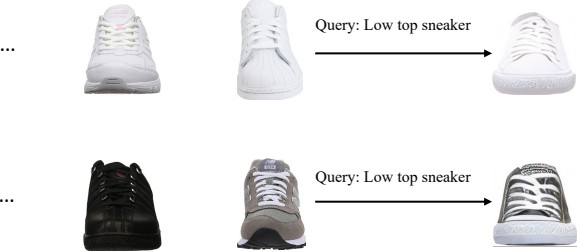

Figure 3: Visualization of the generated images.

In Fig. 3, we present a visualization of the images generated by UniMP. Initially, it is evident that the generated images possess a high degree of realism and clarity. Moreover, when examining the two generated images, it becomes apparent that even when provided with the identical query "low top sneaker," our model has the capability to produce diverse and pertinent images tailored to individual user history.

## 4 Related Work

**Multi-modal Personalization.** The development of multi-modal personalization systems is crucial due to the prevalence of multi-modal information on the Web. The most common usage is to leverage multi-modal content as side information to assist recommendation decisions. For example, VBPR (He & McAuley, 2016b) and AMR (Tang et al., 2019) propose using visual features to support collaborative filtering computation. MMGCN (Wei et al., 2019) designs a multi-modal Graph Convolution network framework, which can yield modal-specific representations of users and micro-videos to better capture user preferences. CausalRec (Qiu et al., 2021) develops a causal inference framework to effectively make use of the visual feature and in the meanwhile remove the visual bias. MMSSL (Wei et al., 2023b) proposes a modality-aware interactive structure learning paradigm via adversarial perturbations to generate robust representations. However, all the above methods only focus on single personalized recommendation tasks and fail to model the mixture of visual, textual, and ID data, arranged in arbitrary sequences.

**General-purpose Model.** Language is an incredibly powerful tool that can effectively describe a wide range of concepts. With the advancements in modern language models, language has evolved into a versatile interface that can be applied to various tasks and modalities. Several methods have emerged to connect different tasks and modalities within a unified language model. VL-T5 (Cho et al., 2021) and Pix2Seq (Chen et al., 2021) approach vision-language tasks and object detection by formulating them as sequence-to-sequence problems based on conditional language generation. To solve the text-to-image generation task, DALL-E (Ramesh et al., 2021), on the other hand, tackles the text-to-image generation task by converting images into latent tokens and employing a Transformer decoder to model both text and image tokens in an autoregressive manner. In a similar vein, OFA (Wang et al., 2022) has contributed to the development of comprehensive vision-language foundation models that excel in generating text from images. Flamingo (Alayrac et al., 2022; Awadalla et al., 2023), BLIP (Li et al., 2023c) and Otter (Li et al., 2023a) further advance multi-modal foundation models, exploring robust modal alignment and instruction tuning mechanisms. Moving towards personalization, P5 (Geng et al., 2022) has proposed a unified framework that integrates various recommendation tasks, while VIP5 (Geng et al., 2023) extends this framework to a multi-modal setting. However, they fail to fully harness the potential of raw data and do not effectively capture the interactions among them. They lack the required flexibility to effectively handle the diverse input and output requirements inherent in multi-task learning.

**Large language model for personalization.** Personalization has been well-studied for information access problems. Recently, large language models (LLMs) have shown great performance in various personalization tasks. Some works explored personalization for fundamental language modeling with the openly available user data on Reddit (Welch et al., 2022), Facebook, and Twitter (Soni et al., 2022). (Soni et al., 2022) also explore applying a personalized language model for downstream tasks in stance classification and demographic inference. Similarly, other work has explored personalized sentiment prediction tasks on publicly available Yelp and IMDB data (Mireshghallah et al., 2022; Zhong et al., 2021). The large language model has demonstrated remarkable success in learning unified representations and has the potential to enhance the personalization of recommendation systems (Geng et al., 2022; Li et al., 2023b; Hou et al., 2022) as well. (Salemi et al., 2023) introduces the LaMP benchmark and highlights the importance of personalization for LLMs. However, most previous methods mainly focus on the text and fail to model the multi-modal data.

## 5 Conclusion

In the paper, we developed a unified data format designed for multi-modal user history integration. Our flexible framework is uniquely positioned to handle multi-modal user inputs and generate corresponding multi-modal outputs tailored to individual needs. Our proposed user modeling architecture is optimized for meticulous multi-modal data extraction and alignment, ensuring accurate predictions of user preferences. Through our multi-task optimization strategy, we further enhanced the generalization prowess of `UniMP`. Additionally, we undertook the initiative to benchmark several multi-modal personalized tasks including recommendation, retrieval, prediction, and generation, catering to a diverse set of user requirements. Comprehensive evaluation affirmed the superiority of `UniMP`, demonstrating its capability to outperform specialized competitor methods across tasks and largely enhance its transferability.

## ACKNOWLEDGEMENT

This work is supported by National Science Foundation under Award No. IIS-2117902, and Agriculture and Food Research Initiative (AFRI) grant no.2020-67021-32799/project accession no.1024178 from the USDA National Institute of Food and Agriculture. The views and conclusions are those of the authors and should not be interpreted as representing the official policies of the funding agencies or the government.

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

# A APPENDIX

The code and model weights will be open-sourced after the review procedure.

## A.1 DATASET STATISTICS

Table 5: Statistics of datasets

|          | ♯ Users | ♯ Items | ♯ Interactions | Sparsity |
|----------|---------|---------|----------------|----------|
| Baby     | 19,822  | 7,776   | 163,856        | 99.89%   |
| Beauty   | 25,837  | 16,893  | 227,920        | 99.95%   |
| Clothing | 58,197  | 44,310  | 422,474        | 99.98%   |
| Grocery  | 16,318  | 11,581  | 165,893        | 99.91%   |
| Sports   | 40,358  | 24,766  | 334,238        | 99.97%   |
| Toys     | 24,314  | 18,906  | 209,281        | 99.95%   |
| Office   | 6,913   | 4,775   | 68,306         | 99.79%   |

Table 5 displays the dataset's statistics. The dataset spans several domains, including Amazon-Baby, Amazon-Beauty, Amazon-Clothing, Amazon-Grocery, Amazon-Sports, Amazon-Toys, and Amazon-Office. Amazon-Office was exclusively set aside to test the model's adaptability to the new domain, while the remaining domains were used for training and in-domain evaluation. Similar to the setting of previous papers (Wei & He, 2022; Wei et al., 2020), only users and items with a minimum of five input interactions were considered for training. For each user, the most recent interaction was used as the test sample, the penultimate for validation, and all prior interactions for training. Additionally, we reserved 10% of users from the training domains to assess the model's generalization for new users during testing.

## A.2 IMPLEMENTATION DETAILS

During model training, we train our model for 10 epochs with a learning rate searched from {2e-4,1e-4,1e-5}. The learning rate is set to 2e-4 by default. We employ the AdamW optimizer, with a warm-up ratio of 0.01 and a gradient accumulation step of 2. For our visual encoder, we use the CLIP ViT-L model (Radford et al., 2021). As for the pre-trained large language model, we employ the instruction-tuned 3B model (Together.ai, 2023), which has already undergone instruction tuning. We intend to explore different language model sizes in future research. The cross-attention module is integrated every 2nd language model layer. Our language model consists of 24 layers with a hidden dimension of 4096. Consistent with Alayrac et al. (2022); Awadalla et al. (2023), the pre-trained language and vision models remain frozen during our process. Only the cross-attention modules, input, and output layers are fine-tuned. Training is conducted with a batch size of 24 on a node with 8 A100 40G GPUs. We employ the cosine learning scheduler for optimal convergence. For visual inputs, we utilize the original images, while textual data includes item category, title, price, ID, and brand. Each item in the corpus will be assigned a unique new token in the token table. Evaluation is based on a user's last five interactions as their history. For search and recommendation task, we use the beam search (Zhou & Hansen, 2005) of size 10 to generate the products. The detailed task information will be introduced later.

We compare with the representative and state-of-the-art baselines of each task. Notably, P5 (Geng et al., 2022) and VIP5 (Geng et al., 2023) are unified models that can work on partial of tasks. For recommendation, we split the baselines into four categories - general recommenders: MF (Koren et al., 2009; Rendle et al., 2012), MACR (Wei et al., 2021), LightGCN (He et al., 2020), Ultra-GCN (Mao et al., 2021). Sequential recommenders: HGN (Ma et al., 2019), GRU4Rec (Jannach & Ludewig, 2017), SASRec (Kang & McAuley, 2018) and S³-Rec (Zhou et al., 2020). Generative recommenders: BERT4Rec (Sun et al., 2019), UniSRec (Hou et al., 2022), P5 (Geng et al., 2022) and VIP5 (Geng et al., 2023). Multi-modal recommenders: VBPR (He & McAuley, 2016b), CausalRec (Qiu et al., 2021), MMGCL (Yi et al., 2022) and MMSSL (Wei et al., 2023b). In addition the recommenders, we use MRG (Truong & Lauw, 2019) as baseline for explaination generation, SANCL Han et al. (2022) for preference prediction, S³-Rec (Zhou et al., 2020) and UniSRec (Hou et al., 2022) utilizing threshold-based method for multi-modal selection, BM25 (Robertson et al.,

2009), HEM (Ai et al., 2017), ZAM (Ai et al., 2019), RTM (Bi et al., 2021). Note that general recommenders operate differently compared to other types and are included primarily to ensure a thorough and comprehensive comparison. It's important to note that the other categories (e.g., generative recommenders) are our primary baselines for comparison. We leave the comparisons with multi-view learning methods (Fu et al., 2020) as our future work.

We propose multiple personalized multi-modal tasks to evaluate the performance of our proposed `UniMP`. Details of the tasks are shown below.

**Personalized recommendation.** In this task, given the user's history information, we want the predict what product the user will prefer next. Combining with the user history, the task instruction is denoted as "Can you recommend the next item to the user?".

**Personalized preference prediction.** In this task, given the user's history information, we want the predict the exact preference (1-5) of the user to each specific product. Given an item $h$ and user history, the task instruction is represented as "Given $i_h$, can you predict the preference level of the user to the product?" where $i_h$ is the contextual information of item $h$ including $i_h = \{[\text{IMG}], k_1, v_1, k_2, v_2, \ldots, k_m, v_m, [\text{EOC}]\}$. The visual information $x_h$ will also be processed and cross-attended by the model.

**Personalized explanation generation.** In this task, given the user's history information and an item, we want to explain why the user may want the product. Given an item $h$, the task instruction is represented as "Given $i_h$, can you explain to the user why the product might be suitable or unsuitable?".

**Personalized multi-modal selection.** In this task, given the user's history information, we want the select the appropriate products for the user from the candidates $\{i_{u_1}, i_{u_2}, \ldots i_{u_e}\}$. The task instruction is represented as "Given $i_{u_1}, i_{u_2}, \ldots i_{u_e}$, can you select all the appropriate items for the user?". In practice, to construct the candidates, we randomly select a varying number (1-3) of positive items along with a diverse set of negative samples.

**Personalized multi-modal search.** In this task, using the user's historical data, we aim to retrieve relevant products in response to the user's query. The task instruction is phrased as: "[Query] Can you retrieve the items that aligns with both the user's history and the given query?". As per Ai et al. (2017; 2019); Bi et al. (2021), we generate the query based on the item's category. It's essential to note that the same query can be associated with hundreds of distinct items. Hence, the user's personalized content is crucial for accurately retrieving the appropriate item.

**Personalized user-guided image generation.** In this task, based on the user's history information, we want the generate the product image for the user given the user's query. We first retrieve the relevant items based on history using the personalized multi-modal search instruction to filter out noise. Then we generate the product figure based on the retrieved products. Combined with the user history, the retrieval instruction is formulated as: "[Query] Can you retrieve the items that aligns with both the user's history and the given query?". With the retrieved images, the image generation instruction is represented as: "[Retrieved Images] [Query] Can you generate the product figure that the user may be interested in?". In this way, users can more easily find and discover items of interest, leading to improved user experience, which supplement users' diverse information needs. The number of retrieved images is set to 2 in the paper.

## A.3 ROBUSTNESS OF PROPOSED METHOD TO MISSING MODALITY (RQ5)

In Table 6, we present the robustness of `UniMP` in the face of noisy textual and visual inputs. Here, NT represents textual input in which 20% of the attribute information is absent, and NV denotes visual input where 20% of the images do not match their corresponding product. Both of these scenarios are reflective of real-world application conditions. It is evident from the table that `UniMP` is robust to such noisy input data, and it will effectively leverages the other modality to deliver accurate predictions with minimal performance loss. On the other hand, the performance of the baseline methods MMSSL and $S^3$-Rec deteriorates considerably in these conditions.

Table 6: Robustness to Textual and Visual Information. NT represents textual input in which partial attribute information is absent, and NV denotes visual input where partial images do not match their corresponding product.

|  | HR@5 | NDCG@5 | MRR@5 |
|---|---|---|---|
| $S^3$-Rec with NT | 0.0275 | 0.0158 | 0.0133 |
| $S^3$-Rec | 0.0311 | 0.0193 | 0.0156 |
| MMSSL with NV | 0.0253 | 0.0171 | 0.0144 |
| MMSSL | 0.0281 | 0.0194 | 0.0164 |
| UniMP with NT | 0.0332 | 0.0228 | 0.0194 |
| UniMP with NV | 0.0328 | 0.0226 | 0.0192 |
| UniMP (Ours) | **0.0337** | **0.0231** | **0.0196** |

Table 7: Analysis of Learning Strategy.

|  | Recommendation | | | Search | | |
|---|---|---|---|---|---|---|
|  | HR@5 | NDCG@5 | MRR@5 | HR@5 | NDCG@5 | MRR@5 |
| Continue | 0.0197 | 0.0162 | 0.0134 | 0.0876 | 0.0672 | 0.0583 |
| Multi-task | 0.0293 | 0.0220 | 0.0196 | 0.1632 | 0.1232 | 0.1105 |
| Multi-task+Full | 0.0325 | 0.0227 | 0.0189 | 0.1654 | 0.1254 | 0.1121 |
| Multi-task+Efficient | **0.0337** | **0.0231** | **0.0196** | **0.1769** | **0.1309** | **0.1158** |

## A.4 ANALYSIS OF LEARNING STRATEGY

In Table 7, we examine various learning strategies, leading to several observations: Continuous learning tends to result in the loss of previously acquired knowledge, leading to subpar performance. Multi-task learning surpasses the performance of continuous learning, indicating the acquisition of transferable task knowledge. However, multi-task learning might not achieve optimal results due to the balancing between different tasks. When we further fine-tuned the model for specific tasks (denoted as Multi-task+Full), we observed that adjusting only a subset of parameters (specifically, the last layer, denoted as Multi-task+Efficient) enhances overall performance more effectively than fine-tuning the entire model. This also motivates future work on designing more effective parameter-efficient fine-tuning strategies for the multi-modal user modeling method.

## A.5 RATIONALE OF DATA FUSION METHOD

Our fusion method's main innovation, compared to CLIP (Radford et al., 2021) and VIP5 (Geng et al., 2023), lies in our unique approach to fusion strategy and fusion position. CLIP processes visual and textual data separately through modality-specific encoders and only combines them at the final layer to compute similarity scores. VIP5, on the other hand, merges visual embeddings, obtained from a frozen visual encoder, with textual inputs right at the beginning for language modeling. We refer to these as "late fusion" and "early fusion" strategies, respectively. These methods focus on processing a single image-text pair.

In contrast, UniMP is designed to utilize a user's entire history, involving multiple products, for the generation. To handle this, we proposed UniMP to align and integrate the visual information into the language model through the specific cross-attention design (instead of concatenation). The visual information of each product is conditioned exclusively on its corresponding textual data during fusion. We found that removing exclusive attention leads to a substantial performance decrease (0.0337→0.0244). This fusion is also performed at every layer, allowing for a more nuanced and detailed understanding of user preferences. To demonstrate the effectiveness of this approach, we've included an experiment titled 'w/o Fine-grained' in Table 3, showing our method's performance without the cross-attention design. Additionally, we have conducted comparisons with Early Fusion, Late Fusion, and without Exclusive Attention strategies respectively in Table 8, which further validate the effectiveness and rationale behind our proposed fusion method.

Table 8: Ablation on Data Fusion Strategies.

|  | HR@5 | NDCG@5 | MRR@5 |
|---|---|---|---|
| VIP5 | 0.0262 | 0.0163 | 0.0127 |
| UniMP w/o Exclusive Attention | 0.0244 | 0.0165 | 0.0132 |
| UniMP w/o Cross-attention | 0.0257 | 0.0192 | 0.0161 |
| UniMP with Early Fusion | 0.0271 | 0.0185 | 0.0135 |
| UniMP with Late Fusion | 0.0274 | 0.0193 | 0.0156 |
| UniMP | 0.0337 | 0.0231 | 0.0196 |

## A.6 NECESSITY OF CONTEXT RECONSTRUCTION AND TOKEN-LEVEL REWEIGHTING

For the context reconstruction loss, this term involves predicting product attributes for each historical product image in a user's sequence. Unlike our primary task in the paper (e.g., recommendation), which utilizes the entire user history encompassing multiple products, context reconstruction loss focuses on individual products. This approach aligns with the pre-training objective of the model, such as generating detailed descriptions for each specific image. The integration of context reconstruction loss serves as a bridge, harmonizing the optimization of our model. Importantly, the inclusion of context reconstruction loss doesn't incur extra computational costs, yet it substantially enhances training effectiveness by enriching the information learned.

As for token-level reweighting, we recognize that reconstructing context is generally simpler than accurately predicting user preferences, which is a key motivation for our approach. To address this, we propose a reweighting term designed to automatically differentiate between tokens that are easier or harder to predict.

## A.7 EMPIRICAL PERFORMANCE ON MORE MULTI-MODAL DATASETS

We primarily selected the Amazon dataset as our experimental platform due to its extensive and varied content, including reviews, ratings, images, interactions, and product attributes. This comprehensiveness makes it particularly suited for multi-modal, multi-task evaluations. Recognizing the importance of validating our findings across different datasets, we have also applied our UniMP method to the Netflix and HM datasets. For Netflix, the images are obtained by crawling movie posters from the website. The results, as detailed in Table 9, affirm the effectiveness and adaptability of UniMP across these varied datasets.

Table 9: Recommendation Results on HM and Netflix Datasets.

|  | HM | | | Netflix | | |
|---|---|---|---|---|---|---|
|  | HR@5 | NDCG@5 | MRR@5 | HR@5 | NDCG@5 | MRR@5 |
| P5 | 0.0101 | 0.0063 | 0.0046 | 0.0742 | 0.0413 | 0.0316 |
| VIP5 | 0.0122 | 0.0118 | 0.0093 | 0.0936 | 0.0589 | 0.0395 |
| S3-Rec | 0.0185 | 0.0121 | 0.0102 | 0.1155 | 0.0632 | 0.0511 |
| UniSRec | 0.0196 | 0.0139 | 0.0107 | 0.1324 | 0.0856 | 0.0644 |
| UniMP (Ours) | 0.0313 | 0.0206 | 0.0172 | 0.1723 | 0.1196 | 0.1024 |

While we are keen to broaden our multi-modal validation to encompass video and audio data, we currently face the limitation of not having access to publicly available datasets containing raw video or audio content. Despite this, our method holds the capability for extension to various other data modalities. For instance, we could process image frames from a video through our vision encoder and apply pooling techniques to derive representations for further analysis.

## A.8 CHOICES OF LANGUAGE MODELS

In our paper, we utilized the 3B INCITE model, an instruction-tuned large language model, for its ease of use and open-source availability. It is important to note that our framework is adaptable to various large language models (LMs). To further explore this adaptability, we tested UniMP's performance with Mosaic ML's MPT-1B model, denoted as UniMP-Small. The results of these

Table 10: Recommendation Results with Different LMs.

|  | HR@5 | NDCG@5 | MRR@5 |
| --- | --- | --- | --- |
| UniMP-Small | 0.0315 | 0.0216 | 0.0179 |
| UniMP | **0.0337** | **0.0231** | **0.0196** |

experiments, detailed in Table 10, suggest that larger LMs can more effectively improve personalized user modeling.

## A.9 FUTURE WORK

For the future work of our model, there's potential to seamlessly integrate active user feedback loops. This would facilitate real-time, accurate evaluations rooted in actual user interactions and preferences. Additionally, expanding the range of incorporated modalities, such as integrating audio data, promises a more holistic user experience. As we strive for better product ID generation, delving into semantic ID strategies (Rajput et al., 2023; Zeng et al., 2023; Jin et al., 2023) becomes crucial. Moreover, the emphasis on transferable learning highlights the importance of researching parameter-efficient transfer mechanisms (Hu et al., 2021; Wei et al., 2023a) in multi-modal user modeling. The development of adaptive algorithms (Ban et al., 2022; Qi et al., 2023; 2024) and decentralized methods (Wu et al., 2024; Bao et al., 2024; Sun et al., 2023) are also crucial for the real-world applications of multi-modal personalized models. Lastly, for content generation tasks, leveraging larger and more diverse datasets for pre-training may further enhance model effectiveness.

