# OpenReview forum: "Towards Unified Multi-Modal Personalization: Large Vision-Language Models for Generative Recommendation and Beyond"
_ICLR.cc/2024/Conference — ICLR 2024 poster_

### Official Review · Reviewer_4rAY · 2023-10-29

**Soundness:** 3 good
**Presentation:** 3 good
**Contribution:** 3 good
**Rating:** 8
**Confidence:** 3

**Summary:**

This study proposes UniMP, a method for enhancing individual user
experiences by integrating multi-modal user information
effectively. It introduces a flexible data format for combining
diverse user inputs, fine-grained user modeling, and a multi-task
optimization approach. UniMP can outperform specialized methods in
various personalization tasks and is particularly effective in
transfer learning scenarios with new users and domains. It also
demonstrates the ability to generate personalized content and handle
noisy multi-modal input. The study highlights the importance of
context reconstruction and token-level re-weighting mechanisms in
improving training effectiveness. Overall, UniMP offers a versatile
approach to multi-modal personalization.

**Strengths:**

1. UniMP seamlessly integrates multi-modal user data, accommodating
various input types, enhancing personalized recommendations.

2. The model's fine-grained user modeling ensures accurate user
preference predictions, improving overall performance.

3. UniMP excels in transfer learning, adapting well to new users and
domains, providing robust personalized recommendations.

**Weaknesses:**

The study's weakness lies in its perceived lack of novelty in
combining image and language learning, which may not be considered
highly innovative.

You claim it's "multi-modal," but it only deals with language and
images. Since there's no validation with video or audio, the term
"universal" might be overstated.

**Questions:**

You claim it's "multi-modal," but it only deals with language and
images. Since there's no validation with video or audio, the term
"universal" might be overstated. Would it be advisable to revise the
title or abstract to specify "language and images" to better reflect
the scope?

The dataset's limitation to Amazon data is acknowledged. To enhance
the study's scope, have you considered validating it with datasets
like Yelp, where there's a combination of image uploads, reviews, and
ratings? Additionally, could you extend the multi-modal validation to
include video and audio data, such as YouTube viewing history?

---

> ### Author Response · Authors · 2023-11-18
> **Response to Reviewer 4rAY (1/2)**
>
> Thank you for your valuable review and suggestions. We deeply appreciate your thoughtful feedback and are grateful for the time and effort you've invested.
>
> > **Q1:** The dataset's limitation to Amazon data is acknowledged.
>
> Thank you for your valuable suggestion. We primarily selected the Amazon dataset as our experimental platform due to its extensive and varied content, including reviews, ratings, images, interactions, and product attributes. This comprehensiveness makes it particularly suited for multi-modal, multi-task evaluations. Recognizing the importance of validating our findings across different datasets, we have also applied our UniMP method to the Netflix [1] and HM [2] datasets. For Netflix, the images are obtained by crawling movie posters from the website. The results, as detailed in our tables, affirm the effectiveness and adaptability of UniMP across these varied datasets.
>
> |              |               |          HM        |                 |          |        Netflix          |                 |
> |:------------:|:---------------:|:----------------:|:---------------:|:---------------:|:----------------:|:---------------:|
> |              |       HR@5      |      NDCG@5      |      MRR@5      |       HR@5      |      NDCG@5      |      MRR@5      |
> |      P5      |      0.0101     |      0.0063      |      0.0046     |      0.0742     |      0.0413      |      0.0316     |
> |     VIP5     |      0.0122     |      0.0118      |      0.0093     |      0.0936     |      0.0589      |      0.0395     |
> |    S3-Rec    |      0.0185     |      0.0121      |      0.0102     |      0.1155     |      0.0632      |      0.0511     |
> |    UniSRec   |      0.0196     |      0.0139      |      0.0107     |      0.1324     |      0.0856      |      0.0644     |
> | UniMP (Ours) | **0.0313** | **0.0206** | **0.0172** | **0.1723** | **0.1196** | **0.1024** |
>
> While we are keen to broaden our multi-modal validation to encompass video and audio data, we currently face the limitation of not having access to publicly available datasets containing raw video or audio content. Despite this, our method holds the capability for extension to various other data modalities. For instance, we could process image frames from a video through our vision encoder and apply pooling techniques to derive representations for further analysis.
>
> > **Q2:** The term "universal" might be overstated. Would it be advisable to revise the title or abstract to specify "language and images" to better reflect the scope?
>
> We appreciate the constructive feedback and valuable suggestions. We have updated the abstract to explicitly mention 'language and images,' ensuring it more accurately represents the scope of our study. Additionally, we've substituted the term 'universal' with 'unified' throughout the paper, as it more appropriately aligns with our work's focus and scope. We will also update the title to reflect these changes as soon as OpenReview enables this functionality.

---

> > ### Author Response · Authors · 2023-11-18
> > **Response to Reviewer 4rAY (2/2)**
> >
> > > **Q3:** Combining image and language learning may not be considered highly innovative.
> >
> > We appreciate the reviewer's assessment. For modal fusion within the network, while bridging pre-trained vision and language models and the concept of cross-attention is not new, our implementation and the specific challenges we address in our paper are unique. In previous works, CLIP [3] processes visual and textual data separately through modality-specific encoders and only combines them at the final layer to compute similarity scores. VIP5 [4], on the other hand, merges visual embeddings, obtained from a frozen visual encoder, with textual inputs right at the beginning for language modeling. We refer to these as "late fusion" and "early fusion" strategies, respectively. These methods focus on processing a single image-text pair. In contrast, UniMP is designed to utilize a user's entire history, involving multiple products, for the generation. To handle this, we proposed UniMP to align and integrate the visual information into the language model through the specific cross-attention design (instead of concatenation). The visual information of each product is conditioned exclusively on its corresponding textual data during fusion. We found that removing exclusive attention leads to a substantial performance decrease (0.0337$\rightarrow$0.0244). This fusion is also performed at every layer, allowing for a more nuanced and detailed understanding of user preferences. To demonstrate the effectiveness of this approach, we've included an experiment titled 'w/o Fine-grained' in Table 3 of our submitted paper, showing our method's performance without the cross-attention design. Additionally, we have conducted comparisons with Early Fusion, Late Fusion, and without Exclusive Attention strategies respectively in the below table, which further validate the effectiveness and rationale behind our proposed fusion method. We provide detailed illustrations of the novelty of our paper in response to Q3 of reviewer s3tt.
> >
> >
> > |                         |     HR@5     |   NDCG@5   |    MRR@5   |
> > |:-----------------------:|:------------:|:----------:|:----------:|
> > |           VIP5          |    0.0262    |   0.0163   |   0.0127   |
> > |  UniMP w/o Exclusive Attention |   0.0244    |   0.0165   |   0.0132  |
> > |  UniMP w/o Cross-attention |   0.0257     |   0.0192   |   0.0161   |
> > | UniMP with Early Fusion |    0.0271    |   0.0185   |   0.0135   |
> > |  UniMP with Late Fusion |   0.0274     |   0.0193   |   0.0156   |
> > |          UniMP          | **0.0337** | **0.0231** | **0.0196** |
> >
> >
> > [1] Netflix Prize data. https://www.kaggle.com/datasets/netflix-inc/netflix-prize-data
> >
> > [2] H&M Personalized Fashion Recommendations. https://www.kaggle.com/competitions/h-and-m-personalized-fashion-recommendations
> >
> > [3] Learning Transferable Visual Models From Natural Language Supervision. ICML 2023
> >
> > [4] VIP5: Towards Multimodal Foundation Models for Recommendation.

---

> > > ### Comment · Reviewer_4rAY · 2023-12-03
> > >
> > > Thank you for your responses. I will maintain my current score.

---

### Official Review · Reviewer_s3tt · 2023-10-31

**Soundness:** 2 fair
**Presentation:** 3 good
**Contribution:** 3 good
**Rating:** 6
**Confidence:** 4

**Summary:**

This paper focuses on multi-modal personalization systems where the input consists not only text and ID's also images. The output is also multi-modal output generation. The authors take advantage of LLMS as multi-modal prompts.

**Strengths:**

This paper has following strengths.
  - Good experimental study and results
  - Consider multi-modal multi-tasks

**Weaknesses:**

- This paper has a lot of things, but lacks of novelty.
- All the components are already in the literature, bridging pre-trained vision and language models is not new, cross attention idea etc.

**Questions:**

- What is x (the visual input of the item), how do you provide it in eq 1? Apart from the special token [IMG]?
- If you are already providing the visual input in user's interaction history, you are using the same image in visual encoder again into cross attention module? (As far as l understand, no visual information is added to the user's history, the features come from visual encoder, if this is correct, l suggest that seperate image from the user's history in Fig 1.)
- What is s in page 3 Section 2.1 line 5?

Typos

Page 2, traditional methods suffers -> suffer
page 4, line 2,  infomration -> information
page 4, Section 2.2, the visual input -> The visual input

---

> ### Author Response · Authors · 2023-11-18
> **Response to Reviewer s3tt (1/2)**
>
> Thank you for your valuable review and suggestions. We appreciate the opportunity to make the following clarifications.
>
> > **Q1:** Good experimental study and results. Consider multi-modal multi-tasks. But lack of novelty. The components are shown in the literature.
>
> We greatly appreciate your recognition of our work, particularly our comprehensive experimental study, the robust results obtained, and our proposal of a multi-modal, multi-task approach. In terms of novelty, we are the first to propose the problem of unified multi-modal personalization for various tasks. Our method addresses the inherent challenges through a comprehensive approach that encompasses three complementary aspects: the formulation of input data, the fusion of modalities in network modeling, and the design of optimization objectives. The novelty of our work lies in the unique solution to each perspective and the seamless integration of these components to formulate an overall effective strategy. Next we will elaborate on how each component has been designed and adapted to create a distinctive approach.
>
> For user input modeling, we are the first to emphasize and verify the feasibility of utilizing raw multi-modal content, as opposed to relying solely on single-modal data or product IDs to represent user sequences. This choice of content acts as a bridge, facilitating transferability and generalization across new domains and users. The comparative analysis of generalization capabilities, presented in Table 4 of our paper, highlights the strengths of our method.
>
> For modal fusion within the network, while bridging pre-trained vision and language models and the concept of cross-attention is not new, our implementation and the specific challenges we address in our paper are unique. In previous works, CLIP [1] processes visual and textual data separately through modality-specific encoders and only combines them at the final layer to compute similarity scores. VIP5 [2], on the other hand, merges visual embeddings, obtained from a frozen visual encoder, with textual inputs right at the beginning for language modeling. We refer to these as "late fusion" and "early fusion" strategies, respectively. These methods focus on processing a single image-text pair. In contrast, UniMP is designed to utilize a user's entire history, involving multiple products, for the generation. To handle this, we proposed UniMP to align and integrate the visual information into the language model through the specific cross-attention design (instead of concatenation). The visual information of each product is conditioned exclusively on its corresponding textual data during fusion. We found that removing exclusive attention leads to a substantial performance decrease (0.0337$\rightarrow$0.0244). This fusion is also performed at every layer, allowing for a more nuanced and detailed understanding of user preferences. To demonstrate the effectiveness of this approach, we've included an experiment titled 'w/o Fine-grained' in Table 3 of our submitted paper, showing our method's performance without the cross-attention design. Additionally, we have conducted comparisons with Early Fusion, Late Fusion, and without Exclusive Attention strategies respectively in the below table, which further validate the effectiveness and rationale behind our proposed fusion method.
>
> |                         |     HR@5     |   NDCG@5   |    MRR@5   |
> |:-----------------------:|:------------:|:----------:|:----------:|
> |           VIP5          |    0.0262    |   0.0163   |   0.0127   |
> |  UniMP w/o Exclusive Attention |   0.0244    |   0.0165   |   0.0132  |
> |  UniMP w/o Cross-attention |   0.0257     |   0.0192   |   0.0161   |
> | UniMP with Early Fusion |    0.0271    |   0.0185   |   0.0135   |
> |  UniMP with Late Fusion |   0.0274     |   0.0193   |   0.0156   |
> |          UniMP          | **0.0337** | **0.0231** | **0.0196** |
>
>
> Furthermore, our approach to enhance multi-task optimization, which focuses on improving alignment and prioritization, coupled with the introduction of a new multi-modal personalization benchmark, provides critical insights into this emerging domain.
>
> In summary, the integration of all these components presents a holistic solution for addressing real-world challenges in achieving unified multi-modal personalization. We believe our paper contributes valuable insights and advancements to the field, and we hope that the reviewer will consider these aspects of novelty and innovation in our work.

---

> > ### Author Response · Authors · 2023-11-18
> > **Response to Reviewer s3tt (2/2)**
> >
> > > **Q2:** What is x (the visual input of the item), how do you provide it in eq 1? Apart from the special token [IMG]?
> >
> >
> > Apologies for any confusion caused. The variable $x$ represents the feature matrix of the visual input, in the shape $\mathbb{R}^{H\times W\times C}$, where $H$, $W$, $C$ represents the height, width, and channel count of the image, respectively. Equation 1 serves as the input for the language model and is formatted in plain text. Note that $x$ is not included in Equation 1. We use the special token [IMG] in **plain text** to denote the image's location. It allows us to establish a correspondence between the textual tokens and the image associated with each product. The key purpose here is to ensure that the visual information of each product is conditioned exclusively on its corresponding textual data during cross-modal attention, a factor critical to the success of our method.
> >
> >
> > > **Q3:** Are you are using the same image in visual encoder again into cross attention module? What is s in page 3 Section 2.1 line 5?
> >
> > As shown in the answer to Q1, we exclusively input the image feature into the visual encoder to derive its representation. Subsequently, this visual representation for each product is conditioned solely on its corresponding textual tokens during the cross-attention fusion process. The [IMG] is a special token in plain text, and we didn't input the image feature to the language model directly. $s$ in page 3 Section 2.1 should be the user’s interaction history $u_h$. We've made adjustments in our paper for clarity.
> >
> > > **Q4:** Typos.
> >
> > Thank you for bringing this to our attention. We have corrected the typos in the paper and will persist in enhancing its overall presentation and structure, with the aim of improving readability.
> >
> > [1] Learning Transferable Visual Models From Natural Language Supervision. ICML 2023
> >
> > [2] VIP5: Towards Multimodal Foundation Models for Recommendation.

---

> > > ### Author Response · Authors · 2023-11-23
> > > **Kind Reminder**
> > >
> > > Dear Reviewer s3tt,
> > >
> > > We wish to express our sincere gratitude once again to you for the valuable contributions and considerate feedback. We would like to gently bring to your attention that the discussion phase between authors and reviewers is nearing completion (within 12 hours).
> > >
> > > Given the inclusion of the new experiments and further clarifications, we kindly inquire whether the reviewers might reconsider the evaluation of our submission. Should you have any further insights to share, we are more than willing to sustain our discussion until the deadline.

---

> ### Comment · Reviewer_s3tt · 2023-11-23
>
> Thanks for your response. You have clarified my concerns and I raised my score accordingly.

---

### Official Review · Reviewer_Y26x · 2023-11-01

**Soundness:** 3 good
**Presentation:** 2 fair
**Contribution:** 2 fair
**Rating:** 5
**Confidence:** 2

**Summary:**

This work proposes a unified personalization generative framework from multi-modal sources with the help of various LMs. Specifically, this work devises a generative personalization framework, UniMP, that can suit many downstream applications, including item recommendation, product search, preference prediction, explanation generation, etc. To achieve this, UniMP devises a universal data format, a user modeling architecture by combining a vision model and a language model, and a token generation framework by integrating multiple personalization learning tasks. Experiments on e-commerce datasets validate the effectiveness of the proposed framework.

**Strengths:**

1. The idea of proposing a unified personalization framework is intriguing and promising in light of the rapid development of various LMs.

2. This work addresses several challenges encountered in multi-task learning when the different backbone pre-trained LMs are in different modalities.

**Weaknesses:**

1. Regarding the technical novelty in data fusion and user modeling, the contributions of this work are not impressive.
In particular, the strategies of data fusion and user modeling are kind of straightforward. The effectiveness is not validated. Please clarify or verify your choice.

2. The presentation of this work should improved. Typos can be found without too much effort.
For instance,

- the abbreviation UniMP is not introduced when it first appears in the Introduction (see the 3rd paragraph on page 2);
- in the 1st paragraph of Section 2.1, "Based on s, ...".

Besides, the notations should be reorganized to make them more readable; e.g., $i_i$ (in Eq. (1)) is confusing.

**Questions:**

1. How do you incorporate the behavioral data, like click, browse, and scroll?

2. How do you define and characterize the multi-modal information? The category, brand, description, and price are all textual inputs; why are they defined as multi-modalities in the paper?

3. What are the choices of backbone LMs?

4. How does a user handle the token limits of the LLMs when s/he aims to apply this framework?

5. How to apply the proposed framework to other domains other than e-commerce?

---

> ### Author Response · Authors · 2023-11-18
> **Response to Reviewer Y26x (1/2)**
>
> Thank you for providing your valuable feedback, and we appreciate the opportunity to make the following clarifications:
>
> > **Q1:** The strategies of data fusion and user modeling are kind of straightforward. The effectiveness is not validated.
>
> Thank you for raising the question. Regarding data fusion, for previous works, CLIP [1] processes visual and textual data separately through modality-specific encoders and only combines them at the final layer to compute similarity scores. VIP5 [2], on the other hand, merges visual embeddings, obtained from a frozen visual encoder, with textual inputs right at the beginning for language modeling. We refer to these as "late fusion" and "early fusion" strategies, respectively. These methods focus on processing a single image-text pair. In contrast, UniMP is designed to utilize a user's entire history, involving multiple products, for the generation. To handle this, we proposed UniMP to align and integrate the visual information into the language model through the specific cross-attention design (instead of concatenation). The visual information of each product is conditioned exclusively on its corresponding textual data during fusion. We found that removing exclusive attention leads to a substantial performance decrease (0.0337$\rightarrow$0.0244). This fusion is also performed at every layer, allowing for a more nuanced and detailed understanding of user preferences. To demonstrate the effectiveness of this approach, we've included an experiment titled "w/o Fine-grained" in Table 3 of our submitted paper, showing our method's performance without the cross-attention design. Additionally, we have conducted comparisons with Early Fusion, Late Fusion, and without Exclusive Attention strategies respectively in the table below, which further validate the effectiveness and rationale behind our proposed fusion method. We have also incorporated the results in Appendix A.6 of the revision.
>
>
> |                         |     HR@5     |   NDCG@5   |    MRR@5   |
> |:-----------------------:|:------------:|:----------:|:----------:|
> |           VIP5          |    0.0262    |   0.0163   |   0.0127   |
> |  UniMP w/o Exclusive Attention |   0.0244    |   0.0165   |   0.0132  |
> |  UniMP w/o Cross-attention |   0.0257     |   0.0192   |   0.0161   |
> | UniMP with Early Fusion |    0.0271    |   0.0185   |   0.0135   |
> |  UniMP with Late Fusion |   0.0274     |   0.0193   |   0.0156   |
> |          UniMP          | **0.0337** | **0.0231** | **0.0196** |
>
>
> In our approach to user modeling, we emphasize the use of raw multi-modal content, as opposed to relying solely on single-modal data or product IDs to represent user sequences. This choice of content acts as a bridge, facilitating transferability and generalization across new domains and users. The comparative analysis of generalization capabilities, presented in Table 4 of our paper, highlights the strengths of our method. Notably, UniMP demonstrates a substantial edge in transfer learning scenarios.
>
> > **Q2:** The presentation of this work should improved.
>
> Thanks for pointing out the typos. We've addressed these corrections and polished the paper. We'll continue to improve the overall presentation and organization of the paper, aiming to enhance its readability.
>
> > **Q3:** How do you incorporate the behavioral data, like click, browse, and scroll?
>
> Thanks for the question. For the Amazon dataset, there are purchase and review behaviors available. In the case of Netflix, it provides data on viewing behavior. To accurately model these behaviors, we incorporate the items that have been purchased or viewed, along with their corresponding attributes, into the user sequence. Moreover, review texts are utilized to represent review behaviors. If additional behavioral signals like clicks, browsing, and scrolling are available, we could assign specific behavior-type tokens to distinguish these various activities and describe the duration of each behavior in a textual format.
>
> > **Q4:** How do you define and characterize the multi-modal information? The category, brand, description, and price are all textual inputs; why are they defined as multi-modalities in the paper?
>
> Apologies for any confusion caused. In our paper, we refer to "multi-modal information" as the combination of textual and visual content. Meanwhile, elements like category, brand, description, and price are classified as "heterogeneous information", which can be represented in a textual format.

---

> > ### Author Response · Authors · 2023-11-18
> > **Response to Reviewer Y26x (2/2)**
> >
> > > **Q5:** What are the choices of backbone LMs?
> >
> > In our paper, we utilized the 3B INCITE model, an instruction-tuned large language model, for its ease of use and open-source availability. It is important to note that our framework is adaptable to various large language models (LMs). To further explore this adaptability, we tested UniMP's performance with Mosaic ML's MPT-1B model, denoted as UniMP-Small. The results of these experiments, detailed in the table below, suggest that larger LMs can more effectively improve personalized user modeling. We have also incorporated the results in Appendix A.9 of the revision.
> >
> > |                         |     HR@5     |   NDCG@5   |    MRR@5   |
> > |:-----------------------:|:------------:|:----------:|:----------:|
> > |          UniMP-Small          |    0.0315    |   0.0216   |   0.0179   |
> > |          UniMP          | **0.0337** | **0.0231** | **0.0196** |
> >
> >
> > > **Q6:** How does a user handle the token limits of the LLMs when s/he aims to apply this framework?
> >
> > Thanks for the insightful question. There are several ways to address the problem. First, we, like prior studies [2,3,..], input the most recent interactions rather than the entire history, as this more accurately reflects current user interests while reducing computational workload. Additionally, the user can consider omitting certain product attributes from the sequence. As evidenced in Table 7 of our paper, our UniMP model demonstrates robustness with incomplete and missing textual and visual data. Furthermore, the problem can be handled from the perspective of language modeling itself. Recent research works [4,5] have shown that pre-trained language models can significantly increase token limits while maintaining performance, achieved by modifying the attention mechanism in a post-hoc manner.
> >
> >
> >
> >
> > > **Q7:** How to apply the proposed framework to other domains other than e-commerce?
> >
> > Thank you for your valuable suggestion. We primarily selected the Amazon dataset as our experimental platform due to its extensive and varied content, including reviews, ratings, images, interactions, and product attributes. This comprehensiveness makes it particularly suited for multi-modal, multi-task evaluations. Recognizing the importance of validating our findings across different datasets, we have also applied our UniMP method to the Netflix [6] and HM [7] datasets. For Netflix, the images are obtained by crawling movie posters from the website. The results, as detailed in the below table, affirm the effectiveness and adaptability of UniMP across these varied datasets. We have also incorporated the results in Appendix A.8 of the revision.
> >
> > |              |               |          HM        |                 |          |        Netflix          |                 |
> > |:------------:|:---------------:|:----------------:|:---------------:|:---------------:|:----------------:|:---------------:|
> > |              |       HR@5      |      NDCG@5      |      MRR@5      |       HR@5      |      NDCG@5      |      MRR@5      |
> > |      P5      |      0.0101     |      0.0063      |      0.0046     |      0.0742     |      0.0413      |      0.0316     |
> > |     VIP5     |      0.0122     |      0.0118      |      0.0093     |      0.0936     |      0.0589      |      0.0395     |
> > |    S3-Rec    |      0.0185     |      0.0121      |      0.0102     |      0.1155     |      0.0632      |      0.0511     |
> > |    UniSRec   |      0.0196     |      0.0139      |      0.0107     |      0.1324     |      0.0856      |      0.0644     |
> > | UniMP (Ours) | **0.0313** | **0.0206** | **0.0172** | **0.1723** | **0.1196** | **0.1024** |
> >
> > Furthermore, our framework is versatile enough to be adapted to various other domains, including Twitter and Instagram, provided that the relevant data is accessible. By analyzing multi-modal posts (image plus textual description) browsed by the user, our method is capable of suggesting personalized content tailored to their preferences.
> >
> > [1] Learning Transferable Visual Models From Natural Language Supervision. ICML 2023
> >
> > [2] VIP5: Towards Multimodal Foundation Models for Recommendation.
> >
> > [3] S$^3$-Rec: Self-Supervised Learning for Sequential Recommendation with Mutual Information Maximization. CIKM 2020
> >
> > [4] LM-Infinite: Simple On-the-Fly Length Generalization for Large Language Models.
> >
> > [5] Efficient Streaming Language Models with Attention Sinks.
> >
> > [6] Netflix Prize data. https://www.kaggle.com/datasets/netflix-inc/netflix-prize-data
> >
> > [7] H&M Personalized Fashion Recommendations. https://www.kaggle.com/competitions/h-and-m-personalized-fashion-recommendations

---

> > > ### Author Response · Authors · 2023-11-23
> > > **Kind Reminder**
> > >
> > > Dear Reviewer Y26x,
> > >
> > > We wish to express our sincere gratitude once again to you for the valuable contributions and considerate feedback. We would like to gently bring to your attention that the discussion phase between authors and reviewers is nearing completion (within 12 hours).
> > >
> > > Given the inclusion of the new experiments and further clarifications, we kindly inquire whether the reviewers might reconsider the evaluation of our submission. Should you have any further insights to share, we are more than willing to sustain our discussion until the deadline.

---

> > > > ### Comment · Reviewer_Y26x · 2023-11-23
> > > >
> > > > Thanks for your response. I prefer to keep my previous scores.

---

### Official Review · Reviewer_Epwj · 2023-11-03

**Soundness:** 3 good
**Presentation:** 3 good
**Contribution:** 2 fair
**Rating:** 6
**Confidence:** 4

**Summary:**

This paper introduces UniMP, a unified framework for multi-modal personalization that seeks to simplify the integration of various data modalities and tasks. It constructs a universal data format that facilitates the incorporation of diverse user historical data. It also presents a cross-attention mechanism that enables multi-modal user modeling. Furthermore, it combines several personalization tasks into a cohesive token generation framework and introduces context reconstruction and token-level reweighting for alignment. The experimental results show that UniMP can outperform competitive baselines on various benchmark tasks.

**Strengths:**

- The motivation to establish a unified paradigm for multi-modal recommendation is good.
- The structure is clear and it is well-written.
- The evaluation is extensive and the experimental results look promising.

**Weaknesses:**

- The rational behind the design choice of its approach is not well-explained.
- The experimental setting needs clarification.
- Whether this method can perform well on other datasets except for Amazon datasets is unknown.

**Questions:**

The paper introduces a unified framework for multi-modal recommendation, which is commendably motivated. Nonetheless, I have concerns regarding the design of its approaches and the evaluation setups, as outlined below:
- The authors introduce a cross-attention mechanism to merge visual and textual data. The rationale for this choice remains ambiguous. It is important to discuss how this approach compares to other vision-language fusion techniques such as CLIP[1] and other multi-modal recommendation systems like VIP5.
- The necessity of the context reconstruction loss is still unclear to me. I wonder the detailed design of this loss item and why it can benefit alignment. And about the token-level reweighting, I think it is important to give more definitions/explanations of easy/hard tokens and how to distinguish.
- In the evaluation part, it is crucial to offer clarity on the experimental settings. Some baseline models like MF and LightGCN operate differently from sequential models like S3Rec and BERT4Rec. Ensuring a clear distinction between these setups is essential to maintain fairness in comparisons.
- The experiments utilize sub-datasets from the Amazon dataset, which might possess similar data distributions. This might limit the generalizability of the results. It would be informative to observe the recommendation model's performance, especially in scenarios of zero/few-shot recommendations, on diverse datasets with visual content.

[1] Radford et al. Learning Transferable Visual Models From Natural Language Supervision. ICML 2021.

---

> ### Author Response · Authors · 2023-11-18
> **Response to Reviewer Epwj (1/2)**
>
> Thank you for your valuable review and suggestions. We hope the following answers can address your concerns.
>
> > **Q1:** The rational behind the cross-attention design of its approach is not well-explained. It is important to discuss how this approach compares to other vision-language fusion techniques such as CLIP[1] and other multi-modal recommendation systems like VIP5.
>
> Thank you for bringing out the question. Our fusion method's main innovation, compared to CLIP [1] and VIP5 [2], lies in our unique approach to fusion strategy and fusion position. CLIP processes visual and textual data separately through modality-specific encoders and only combines them at the final layer to compute similarity scores. VIP5, on the other hand, merges visual embeddings, obtained from a frozen visual encoder, with textual inputs right at the beginning for language modeling. We refer to these as "late fusion" and "early fusion" strategies, respectively. These methods focus on processing a single image-text pair.
>
> In contrast, UniMP is designed to utilize a user's entire history, involving multiple products, for the generation. To handle this, we proposed UniMP to align and integrate the visual information into the language model through the specific cross-attention design (instead of concatenation). The visual information of each product is conditioned exclusively on its corresponding textual data during fusion. We found that removing exclusive attention leads to a substantial performance decrease (0.0337$\rightarrow$0.0244). This fusion is also performed at every layer, allowing for a more nuanced and detailed understanding of user preferences. To demonstrate the effectiveness of this approach, we've included an experiment titled "w/o Fine-grained" in Table 3 of our submitted paper, showing our method's performance without the cross-attention design. Additionally, we have conducted comparisons with Early Fusion, Late Fusion, and without Exclusive Attention strategies respectively in the table below, which further validate the effectiveness and rationale behind our proposed fusion method. We have also incorporated the results in Appendix A.6 of the revision.
>
> |                         |     HR@5     |   NDCG@5   |    MRR@5   |
> |:-----------------------:|:------------:|:----------:|:----------:|
> |           VIP5          |    0.0262    |   0.0163   |   0.0127   |
> |  UniMP w/o Exclusive Attention |   0.0244    |   0.0165   |   0.0132  |
> |  UniMP w/o Cross-attention |   0.0257     |   0.0192   |   0.0161   |
> | UniMP with Early Fusion |    0.0271    |   0.0185   |   0.0135   |
> |  UniMP with Late Fusion |   0.0274     |   0.0193   |   0.0156   |
> |          UniMP          | **0.0337** | **0.0231** | **0.0196** |
>
>
> > **Q2:** The necessity of the context reconstruction loss is still unclear to me. I wonder the detailed design of this loss item and why it can benefit alignment. And about the token-level reweighting, I think it is important to give more definitions/explanations of easy/hard tokens and how to distinguish.
>
> We apologize for the missing details. For the context reconstruction loss, this term involves predicting product attributes for each historical product image in a user's sequence. Unlike our primary task in the paper (e.g., recommendation), which utilizes the entire user history encompassing multiple products, context reconstruction loss focuses on individual products. This approach aligns with the pre-training objective of the model, such as generating detailed descriptions for each specific image. The integration of context reconstruction loss serves as a bridge, harmonizing the optimization of our model. Importantly, the inclusion of context reconstruction loss doesn't incur extra computational costs, yet it substantially enhances training effectiveness by enriching the information learned.
>
> As for token-level reweighting, we recognize that reconstructing context is generally simpler than accurately predicting user preferences, which is a key motivation for our approach. To address this, we propose a reweighting term designed to automatically differentiate between tokens that are easier or harder to predict. We've also included the details and discussion about the loss in Section A.7 of the paper.

---

> > ### Author Response · Authors · 2023-11-18
> > **Response to Reviewer Epwj (2/2)**
> >
> > > **Q3:** The experimental setting needs clarification. In the evaluation part, it is crucial to offer clarity on the experimental settings. Some baseline models like MF and LightGCN operate differently from sequential models like S3Rec and BERT4Rec. Ensuring a clear distinction between these setups is essential to maintain fairness in comparisons.
> >
> > Thank you for your question. For the recommendation task, we categorize the baselines into four distinct groups: general recommenders, sequential recommenders, generative model-based recommenders, and multi-modal recommenders. Within these, both MF and LightGCN are classified under general recommenders. These general recommenders operate differently compared to other types and are included primarily to ensure a thorough and comprehensive comparison. It's important to note that the other categories are our primary baselines for comparison. To clarify our methodology and avoid any confusion, we've included additional details about our experimental setup in Section A.2 of the paper.
> >
> > > **Q4:** Whether this method can perform well on other datasets except for Amazon datasets is unknown.
> >
> > Thank you for your valuable suggestion. We primarily selected the Amazon dataset as our experimental platform due to its extensive and varied content, including reviews, ratings, images, interactions, and product attributes. This comprehensiveness makes it particularly suited for multi-modal, multi-task evaluations. Recognizing the importance of validating our findings across different datasets, we have also applied our UniMP method to the Netflix [3] and HM [4] datasets. For Netflix, the images are obtained by crawling movie posters from the website. The results, as detailed in our tables, affirm the effectiveness and adaptability of UniMP across these varied datasets. We have also included the results in Section A.8 in our revision.
> >
> > |              |               |          HM        |                 |          |        Netflix          |                 |
> > |:------------:|:---------------:|:----------------:|:---------------:|:---------------:|:----------------:|:---------------:|
> > |              |       HR@5      |      NDCG@5      |      MRR@5      |       HR@5      |      NDCG@5      |      MRR@5      |
> > |      P5      |      0.0101     |      0.0063      |      0.0046     |      0.0742     |      0.0413      |      0.0316     |
> > |     VIP5     |      0.0122     |      0.0118      |      0.0093     |      0.0936     |      0.0589      |      0.0395     |
> > |    S3-Rec    |      0.0185     |      0.0121      |      0.0102     |      0.1155     |      0.0632      |      0.0511     |
> > |    UniSRec   |      0.0196     |      0.0139      |      0.0107     |      0.1324     |      0.0856      |      0.0644     |
> > | UniMP (Ours) | **0.0313** | **0.0206** | **0.0172** | **0.1723** | **0.1196** | **0.1024** |
> >
> > [1] Learning Transferable Visual Models From Natural Language Supervision. ICML 2023
> >
> > [2] VIP5: Towards Multimodal Foundation Models for Recommendation.
> >
> > [3] Netflix Prize data. https://www.kaggle.com/datasets/netflix-inc/netflix-prize-data
> >
> > [4] H&M Personalized Fashion Recommendations. https://www.kaggle.com/competitions/h-and-m-personalized-fashion-recommendations

---

> > > ### Author Response · Authors · 2023-11-23
> > > **Kind Reminder**
> > >
> > > Dear Reviewer Epwj,
> > >
> > > We wish to express our sincere gratitude once again to you for the valuable contributions and considerate feedback. We would like to gently bring to your attention that the discussion phase between authors and reviewers is nearing completion (within 12 hours).
> > >
> > > Given the inclusion of the new experiments and further clarifications, we kindly inquire whether the reviewers might reconsider the evaluation of our submission. Should you have any further insights to share, we are more than willing to sustain our discussion until the deadline.

---

> > > > ### Comment · Reviewer_Epwj · 2023-11-23
> > > > **Thank you for the update.**
> > > >
> > > > Thank you for the detailed response and additional experiments. It helped clarify many of my concerns. I raised my score accordingly. About my concern with the baseline settings, I suggest adding the categories explicitly in the Table 1, which can be more clear.

---

> > > > > ### Author Response · Authors · 2023-11-23
> > > > > **Thank you for the response**
> > > > >
> > > > > Thank you for your valuable feedback. We'll add the categories explicitly in Table 1 in the next revision. We're glad that we've addressed many of your concerns. Your acknowledgment through an increased score is greatly appreciated, and we remain dedicated to continuous improvements.

---

### Author Response · Authors · 2023-11-21
**General Response**

We thank all the reviewers for their thoughtful comments. We've corrected phrases, word choices, and typos in the paper, and have incorporated the suggestions and experiment results in the revision. Thank you for pointing out the problem and we apologize for any inconvenience and misunderstanding it may have caused.

Here we would like to highlight our responses to the common concerns. We have also carefully considered each raised concern and made the responses to each reviewer.

> Evaluation beyond Amazon Datasets.

We primarily selected the Amazon dataset as our experimental platform due to its extensive and varied content, including reviews, ratings, images, interactions, and product attributes. This comprehensiveness makes it particularly suited for multi-modal, multi-task evaluations. Recognizing the importance of validating our findings across different datasets, we have also applied our UniMP method to the Netflix and HM datasets. For Netflix, the images are obtained by crawling movie posters from the website. The detailed results are shown in the response to the reviewer and affirm the effectiveness and adaptability of UniMP across these varied datasets. We have also included the results in Section A.8 of our revision.

> Cross-Attention Design Rationale.

Our cross-attention fusion method's main innovation lies in our unique approach to fusion strategy and fusion position. CLIP processes visual and textual data separately through modality-specific encoders and only combines them at the final layer to compute similarity scores. VIP5, on the other hand, merges visual embeddings, obtained from a frozen visual encoder, with textual inputs right at the beginning for language modeling. We refer to these as "late fusion" and "early fusion" strategies, respectively. These methods focus on processing a single image-text pair.

In contrast, UniMP is designed to utilize a user's entire history, involving multiple products, for the generation. To handle this, we proposed UniMP to align and integrate the visual information into the language model through the specific cross-attention design (instead of concatenation). The visual information of each product is conditioned exclusively on its corresponding textual data during fusion. We found that removing exclusive attention leads to a substantial performance decrease. This fusion is also performed at every layer, allowing for a more nuanced and detailed understanding of user preferences. To demonstrate the effectiveness of this approach, we've included an experiment titled "w/o Fine-grained" in Table 3 of our submitted paper, showing our method's performance without the cross-attention design. Additionally, we have conducted comparisons with Early Fusion, Late Fusion, and without Exclusive Attention strategies respectively in the response to reviewers below, which further validate the effectiveness and rationale behind our proposed fusion method. We have also incorporated the results in Appendix A.6 of the revision.

> Overall Novelty Clarification.

In terms of novelty, we first propose the problem of unified multi-modal personalization for various tasks and introduce a unified multi-modal personalization benchmark. Our method addresses the inherent challenges through a comprehensive approach that encompasses three complementary aspects: the formulation of input data, the fusion of modalities in network modeling, and the design of optimization objectives. The novelty of our work lies in the unique solution to each perspective and the seamless integration of these components to formulate an overall effective strategy. We've elaborated on how each component has been designed and adapted to create a distinctive approach in both the response to reviewer below and Appendix A.10.

We also want to express our gratitude to the reviewers for recognizing the strong points of our paper, including:
- The motivation to establish a unified personalization paradigm for multi-modal tasks is promising and intriguing. (Reviewer Epwj, Y26x)
- The paper seamlessly integrates multi-modal user data, accommodating various input types, and enhancing personalized tasks. Considering multi-modal multi-tasking is innovative. (Reviewer s3tt, 4rAY)
- The structure is clear and it is well-written. (Reviewer Epwj)
- The evaluation is extensive and the experimental results are promising. The model excels in transfer learning, providing robust personalized recommendations. (Reviewer Epwj, s3tt, 4rAY)
- Practical solutions to address several challenges encountered in multi-task learning and user preference predictions. (Reviewer Y26x, 4rAY)

In closing, we thank the reviewers again for their valuable time and insightful feedback. Since the discussion period is going to end soon, we gently ask for a retrospect regarding our rebuttal and this will allow us to answer any additional questions or concerns you may have before the discussion period ends.

---

### Author Response · Authors · 2023-11-22
**Gentle Reminder**

Dear Reviewers,

We are following up on our rebuttal since the discussion period is going to end in roughly 1 day. We have carefully considered the raised questions, made our responses as below, and incorporated the clarifications in the revision. We kindly request your feedback on our rebuttal, as this will enable us to address any further questions or concerns you might have before the end of the discussion period. We hope that our response has adequately addressed your concerns, and we would be grateful if you could reconsider the rating of our paper.

Thank you very much for your time and efforts spent on the review!

---

### Meta-Review · Area_Chair_4qjE · 2023-12-14

**Metareview:**

This paper introduces UniMP, a unified framework for multi-modal personalization that incorporates: 1) a universal data format to take into account multi-modal user historical data; 2) a cross-attention mechanism that enables multi-modal user modeling; 3) a unified token generation framework to combine several personalization tasks. The experimental results show that UniMP can outperform competitive baselines on various benchmark tasks.

Strength: a framework combines many different ideas for multi-modal personalization. Promising experimental results.

Weakness: limited novelty with all the components studied in the framework already proposed in the literature, without many insights gained from the experiments.

SAC edit: Overall, strong comprehensive results carry this paper, which contributes to our knowledge without needing to innovate on the algorithmic side. I am thus recommending accept.

**Justification For Why Not Higher Score:**

The work combines many existing components, e.g., data fusion, cross-attention, unified generation framework to combine personalization tasks, in the literature for multi-modal personalization. The novelty of the work is thus limited. Although the experiments show promise, not many new insights are gained through the experimentations.

**Justification For Why Not Lower Score:**

N/A

---

### Decision · Program_Chairs · 2024-01-16

Accept (poster)